# Cigarette smoke increases susceptibility to infection in lung epithelial cells by upregulating caveolin-dependent endocytosis

Parker F. Duffney[1¤a], A. Karim Embong[2], Connor C. McGuire[1], Thomas H. Thatcher[3¤b], Richard P. Phipps[1,2,3,4], Patricia J. Sime[1,3,4¤b] *

**1** Department of Environmental Medicine, University of Rochester School of Medicine and Dentistry, Rochester, NY, United States of America, **2** Department of Microbiology and Immunology, University of Rochester School of Medicine and Dentistry, Rochester, NY, United States of America, **3** Lung Biology and Disease Program, University of Rochester School of Medicine and Dentistry, Rochester, NY, United States of America, **4** Division of Pulmonary and Critical Care Medicine, University of Rochester School of Medicine and Dentistry, Rochester, NY, United States of America

¤a Current address: Curriculum in Toxicology and Environmental Medicine, University of North Carolina at Chapel Hill, Chapel Hill, North Carolina, United States of America
¤b Current address: Department of Internal Medicine, Virginia Commonwealth University, Richmond, Virginia, United States of America
* Patricia.Sime@vcuhealth.org

**Data Availability Statement:** All relevant data are within the manuscript and its Supporting Information files.

## Abstract

Cigarette smoke exposure is a risk factor for many pulmonary diseases, including Chronic Obstructive Pulmonary Disease (COPD). Cigarette smokers are more prone to respiratory infections with more severe symptoms. In those with COPD, viral infections can lead to acute exacerbations resulting in lung function decline and death. Epithelial cells in the lung are the first line of defense against inhaled insults such as tobacco smoke and are the target for many respiratory pathogens. Endocytosis is an essential cell function involved in nutrient uptake, cell signaling, and sensing of the extracellular environment, yet, the effect of cigarette smoke on epithelial cell endocytosis is not known. Here, we report for the first time that cigarette smoke alters the function of several important endocytic pathways in primary human small airway epithelial cells. Cigarette smoke exposure impairs clathrin-mediated endocytosis and fluid phase macropinocytosis while increasing caveolin mediated endocytosis. We also show that influenza virus uptake is enhanced by cigarette smoke exposure. These results support the concept that cigarette smoke-induced dysregulation of endocytosis contributes to lung infection in smokers. Targeting endocytosis pathways to restore normal epithelial cell function may be a new therapeutic approach to reduce respiratory infections in current and former smokers.

## Introduction

Cigarette smoke exposure is projected to cause over 8 million deaths worldwide by 2030 [1] and is a risk factor for many diseases such as COPD. Those exposed to cigarette smoke are at

**Funding:** This work was funded in part by NIH grants R01HL120908, T32HL066988, and P30ES001247, the C.Jane Davis and C. Robert Davis Professorship (to PJS); and the Wright Family Professorship (to RPP). This project described in this publication was supported by the University of Rochester CTSA award number UL1TR002001 from the National Center for Advancing Translational Sciences of the National Institutes of Health. The content is solely the responsibility of the authors and does not necessarily represent the official views of the National Institutes of Health.

**Competing interests:** Thomas H Thatcher is a member of the PLOS ONE editorial board. This does not alter the authors' adherence to all the PLOS ONE policies on sharing data and materials. No other authors have competing interests to declare.

an increased risk for viral and bacterial infections in the lung [2, 3]. Respiratory infections can trigger acute exacerbations of COPD symptoms that are responsible for the bulk of the COPD-related morbidity and mortality. We have previously shown that cigarette smoke impairs TLR3 cleavage in primary human small airway epithelial cells resulting in impaired antiviral responses [4]. Similarly, smoke-exposed mice have increased infectivity with influenza A virus [5]. Yet, how cigarette smoke affects epithelial cell function is still unclear.

Endocytosis is an evolutionarily conserved process that allows cells to take up nutrients and sense the extracellular space. Cells use endocytosis to take up pathogen-associated molecular patterns (PAMP) and other signaling molecules [6, 7]. Many studies have highlighted the importance of endocytosis in the activation of receptor signaling [8, 9]. Despite its essential role in cell homeostasis, many respiratory pathogens utilize endocytic pathways to gain entry into the host cell. For example, influenza A virus (IAV) inserts its viral genome into host cells in a pH-dependent process that occurs following endocytic uptake [10, 11]. Lung epithelial cells are the first line of defense against inhaled pathogens and are the target for many respiratory viruses, thus, the dysregulation of endocytosis in these cells may affect viral susceptibility. However, the effect of cigarette smoke on endocytosis in lung epithelial cells is unknown. Here, we investigated the effect of cigarette smoke on several common endocytic pathways in human small airway epithelial cells that are exploited by viruses to gain access to the host cell.

The most well characterized endocytic pathways are characterized by the involvement specific endosome forming proteins, namely clathrin or caveolin. Both clathrin-mediated and caveolin-mediated endocytosis are involved in receptor signaling and protein internalization [12, 13]. However, other forms of endocytosis exist that are independent of clathrin or caveolin. For example, fluid-phase endocytosis is involved in uptake of fluid from the extracellular space and can be critical for immune sensing of the extracellular environment. Macropinocytosis, a common form of fluid phase endocytosis, can be utilized by pathogens to enter target cells [14, 15].

Here, we investigate for the first time, the effect of cigarette smoke exposure on the uptake of various ligands including the dsRNA viral mimetic poly I:C and influenza virus. We show that smoke differentially effects endocytic pathways, increasing caveolin mediated uptake while inhibiting clathrin mediated uptake. Alterations of endocytosis correlated to increased infectivity with IAV in smoke-exposed small airway epithelial cells (SAEC). Given that smoke impairs many facets of innate host defense, the dysregulation of endocytosis may contribute to the increased infectivity seen in smokers. Targeting endocytosis can provide a new therapeutic strategy to combat infection in the lungs of smokers.

## Materials and methods

### Cell culture and reagents

Primary human SAEC were purchased from Lonza (Allendale, NJ) and grown in small airway epithelial growth media with supplements as recommended by the supplier. SAEC were obtained from three different non-smoking donors (Lot 0000203964, a 51 year old male; lot 0000206158, a 58 year old female; lot 0000105938, a 61 year old male). High molecular weight poly I:C was purchased either unconjugated or conjugated to Fluorescein (Invivogen, San Diego, CA). Fluorescein isothiocyanate (FITC) and Alexa Fluor (AF)594 conjugated Choleratoxin B subunit (CtxB), AF488 and AF594 conjugated bovine serum albumin (BSA), AF488 conjugated to human serum transferrin (Tfn), boron-dipyrromethene (BODIPY) labeled lactosylceraminde (LacCer), and FITC conjugated 3000 MW Dextran were purchased from Molecular Probes (Eugene, OR). FilipinIII was purchased from Sigma (St. Louis, MO).

## Cigarette smoke exposure and treatment with fluorescent ligands

Small airway epithelial cells were cultured at the air-liquid interface and exposed to whole cigarette smoke for 60 min as previously described [4]. Following smoke exposure, media in the basal compartment was replaced to remove any residual smoke components. The apical surface remained unwashed. Cells were then treated on the apical surface with FITC labeled poly I:C (0.5 or 2.5μg/mL as indicated in figure legends), AF488-BSA (50μg/mL), AF488 or AF594-CtxB (10μg/mL), BODIPY-LacCer (1uM), AF488-Tfn (125μg/mL) or FITC-Dextran (250μg/mL) for the indicated times. Due to the rapid integration of AF488-CtxB, BODIPY LacCer, and AF488-Tfn, uptake of these molecules was measured essentially as described [16]. Briefly, cells were rested 5 hours for flow following smoke exposure and subsequently loaded with AF488-CtxB, BODIPY-LacCer, or AF488-Tfn for 30 min at 4˚C to allow for binding while slowing endocytosis. Unbound ligand was then washed away with phosphate buffered saline (PBS) and cells were incubated for 30 min at 37˚C to allow for endocytosis to occur. Following incubation with fluorescent ligands, cells were washed 3 times with PBS.

## Flow cytometry

For flow cytometric analysis, cells were treated with fluorescent ligands as described above, fixed in 2% PFA for 10 min, washed with PBS and analyzed on an LSR II 12 color flow cytometer (BD biosciences, San Jose, CA). Experiments were performed in SAEC from 3 donors and at least 5000 cells/treatment were analyzed.

## Imaging and cell counting

Live cell imaging was done on cells treated as described above and incubated with ligand for 6 hours. For time course analysis, separate cultures were used for each time point and unbound ligand was washed away immediately prior to live cell imaging with an Axio Observer.A1 (Zeiss, Oberkochen, Germany) inverted microscope with an Axiocam MRm camera (Zeiss). For confocal imaging of AF594-BSA and AF594-CtxB, cells were rested for 2 hours (BSA) or overnight (CtxB) after smoke exposure and loaded with ligand for 30 min at 4˚C before incubation for 30 min at 37˚C before washing and fixing cells. Inserts were fixed with 4% paraformaldehyde (PFA) for 10 minutes and mounted with prolong diamond with DAPI (Moloecular Probes) and imaged on an Axio imagur.Z1 florescent microscope using an Axiocam HRc camera (Zeiss) or on a FV1000 Olympus CLSM confocal microscope. For image counting, 2–3 images were taken from each of three replicate inserts per condition. Following acquisition, cells were counted either by using Fiji Multi-Count Tool (http://fiji.sc/) or by using SpotDetector module on Icy imaging software (https://icy.bioimageanalysis.org). The percentage of internalization was determined by dividing the fluorescent positive spots by the number of DAPI positive nuclei and multiplying by 100.

## Western blot and antibodies

Cells were washed with PBS prior to harvest and lysed in 50mM Tris 150mM NaCl 1% NP-40, 50mM Tris, 1mM EDTA. Protein was quantified using the bicinchoninic acid assay and 10μg of protein was analyzed by Western blot. Rabbit monoclonal antibodies for caveolin-1 and clathrin heavy chain were purchased from Cell Signaling (3267 diluted 1:5000 and 2413 diluted 1:1000). Mouse anti-GAPHD monoclonal antibody was purchased from Abcam (8245 diluted 1:10,000).

## Cytokine production

Levels of interferon gamma inducible protein 10 (IP-10), interleukin 6 (IL-6), and interleukin 8 (IL-8) in the culture supernatant were determined by ELISA. Interferon activity in the culture supernatants was determined using an ISRE reporter cell line as described previously [4].

## Virus infection

Virus infection was performed essentially as described [4]. Briefly, SAEC were exposed to air or cigarette smoke for 60 min as described above and then rested for 5 hours before infection with Influenza A/WSN/1933 H1N1 (IAV WSN) virus at the indicated MOI for 1 hour. Unbound virus was washed away with PBS and cells were incubated for 24 hours and then fixed with 100% methanol for 10 min. Culture inserts were then removed and blocked with 1% goat serum in 2.5% BSA for 1 hour. Inserts were incubated with a mouse monoclonal antibody to the IAV nucleoprotein (NP) (kindly provided by Dr. Luis Martinez-Sobrido) overnight and detected with goat anti mouse AF488 secondary antibody (Molecular Probes) and mounted with ProLong Diamond with DAPI (Molecular Probes).

## Results

We have previously shown that cigarette smoke impaired the anti-viral responses in primary human SAEC following treatment with poly I:C [4]. Poly I:C is recognized by the pattern recognition receptor (PRR) toll-like receptor 3 (TLR3). TLR3 is located primarily in the endosome, and endocytosis of poly I:C is required for proper antiviral signaling [17]. While we demonstrated that cigarette smoke impairs TLR3 cleavage, which is important for antiviral signaling [4], the effect of cigarette smoke on uptake of poly I:C in SAEC is unknown. Here, we investigated whether smoke altered uptake of fluorescently labeled poly I:C. We first confirmed that fluorescently labeled poly I:C induced a similar antiviral response as unlabeled poly I:C in SAEC. We found that both labeled and unlabeled poly I:C induced production of IP-10, IL-6 and IL8 in air exposed cells (Fig 1A–1C). Similarly, interferon bioactivity of culture supernatants increased with stimulation with either fluorescently labeled or unlabeled poly I:C (Fig 1D). Smoke-exposed SAECs treated with either fluorescently labeled or unlabeled poly I:C had impaired production of IP-10, IL-6, IL-8, and interferons, consistent with our prior report [4] (Fig 1).

To investigate the effect of cigarette smoke on the uptake of poly I:C, SAEC were exposed to air or smoke followed by treatment with fluorescent poly I:C, and separate samples were imaged at various times post-treatment. At 2 hours post treatment, there was no difference in the number of fluorescently labeled cells in smoke and air exposed SAEC. Interestingly, smoke-exposed SAEC had taken more fluorescent poly I:C by 6 and 19 hours post-treatment (Fig 2A). This was in contrast to the finding that smoke exposure impairs production of inflammatory and antiviral mediators (Fig 1). It was apparent that cigarette smoke-exposed cells had increased uptake of fluorescent poly I:C both on the surface as well as within the cell (Fig 2B). Similarly, flow cytometry analysis revealed a higher mean florescent intensity (MFI) in smoke-exposed SAEC treated with fluorescent poly I:C compared to air-exposed, fluorescent poly I:C treated cells (Fig 2C).

We next investigated whether increased uptake was due to altered endocytosis. Though there are many endocytosis pathways, one of the most well described pathways is characterized by the involvement of the adaptor protein clathrin. Clathrin mediated endocytosis (CME) occurs in all cells and is important in cellular signaling [18, 19]. To investigate the effects of cigarette smoke exposure on CME, we treated air and smoke-exposed SAEC with fluorescent BSA, which has been shown to be taken up by CME [20]. We found that BSA fluorescence was

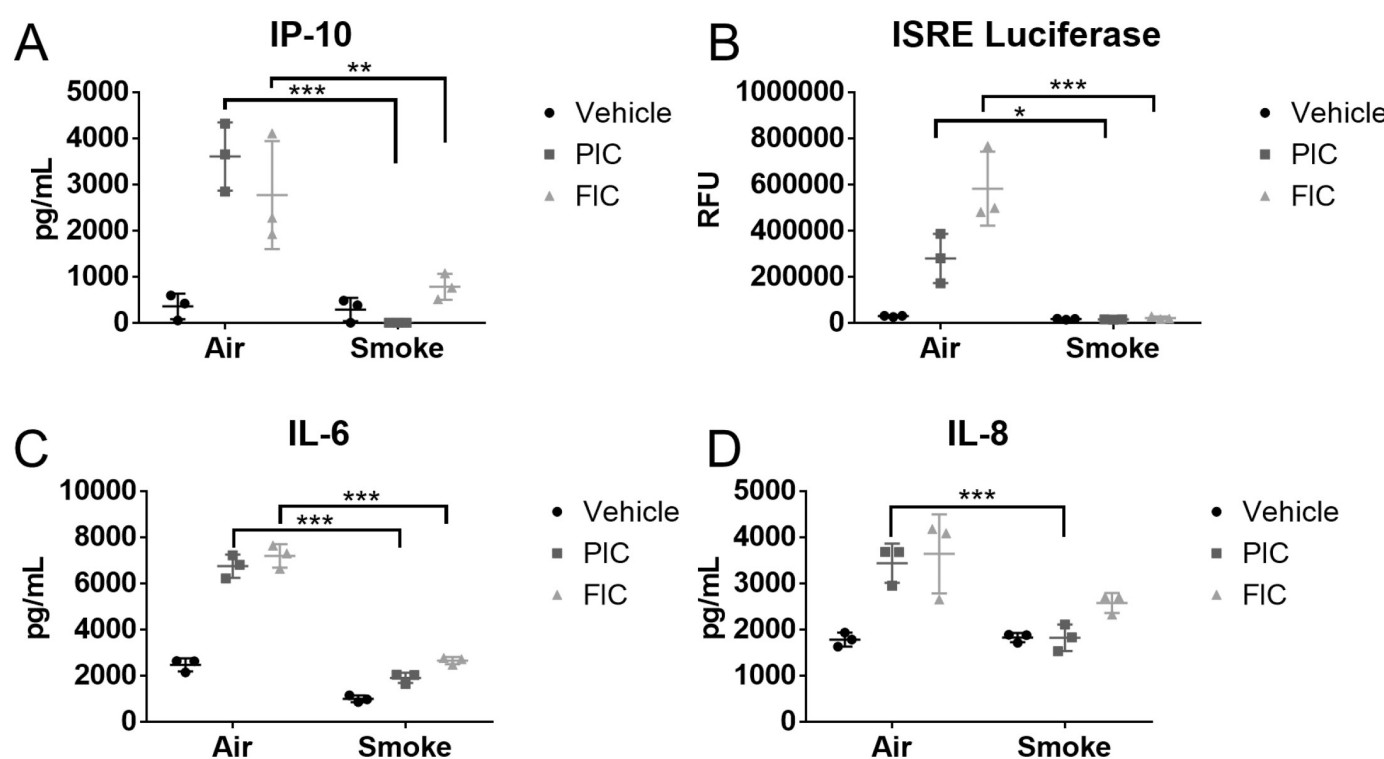

**Fig 1. Fluorescently labeled poly I:C acts similarly to unlabeled poly I:C.** Primary human SAEC were exposed to either air or cigarette smoke for 60 minutes at the air-liquid interface. Following exposure, the basal medium was changed, and the apical surface was treated with 0.5µg/mL of unlabeled poly I:C (PIC, squares) or fluorescent poly I:C (FIC, triangles). Levels of (A) IP-10, (C) IL-6, and (D) IL-8 were measured by ELISA in the culture supernatants 24 hours after treatment with poly I:C. (B) Interferon bioactivity in SAEC supernatants was determined by applying supernatants to reporter cell line expressing luciferase under an interferon reporter as described in the Methods. Bars represent mean ± SD. * p<0.05 **p<0.01 ***p<0.001 by ANOVA with Tukey post hoc correction.

readily detected in air exposed SAEC (Fig 3A and 3B). Unlike poly I:C, smoke-exposed SAEC had reduced uptake of fluorescent BSA (Fig 3A and 3B). To confirm this finding, we quantified the fluorescence of BSA treated cells by flow cytometry and found that the mean fluorescence intensity (MFI) of fluorescent BSA treated cells was reduced with previous cigarette smoke exposure (Fig 3C). Similarly, uptake of AF488-transferrin, another CME ligand, was decreased in smoke-exposed SAEC (Fig 3D).

Another well-described endocytic pathway involves the protein caveolin. Caveolin mediated enodicytosis (CavME) is highly regulated and is dependent on membrane cholesterol [21]. To investigate the effect of cigarette smoke on CavME, we measured uptake of a fluorescently labeled cholera toxin B (CtxB), a ligand that is known to bind cell surface glycolipids and be taken up by CavME [22]. Unlike BSA and transferrin (Tfn), there was little uptake of AF594-CtxB in air exposed SAEC, however, there was a significant increase in the number of cells taking up CtxB after smoke exposure (Fig 4A and 4B). Flow cytometry confirmed an increased MFI of smoke-exposed cells after treatment with AF488-CtxB compared to air exposed cells (Fig 4C). Similarly, uptake of BODIPY-LacCer, another CavME dependent ligand, was similarly increased in smoke-exposed SAEC (Fig 4D).

While the effects of cigarette smoke on CME and CavME are striking, other endocytic pathways are independent of clathrin and caveolin [23]. For example, macropinocytosis is involved in fluid phase endocytosis in which cells sample extracellular fluid [24]. Like CME and CavME, this process can be utilized by pathogens to gain entry into host cells [14]. Thus, we

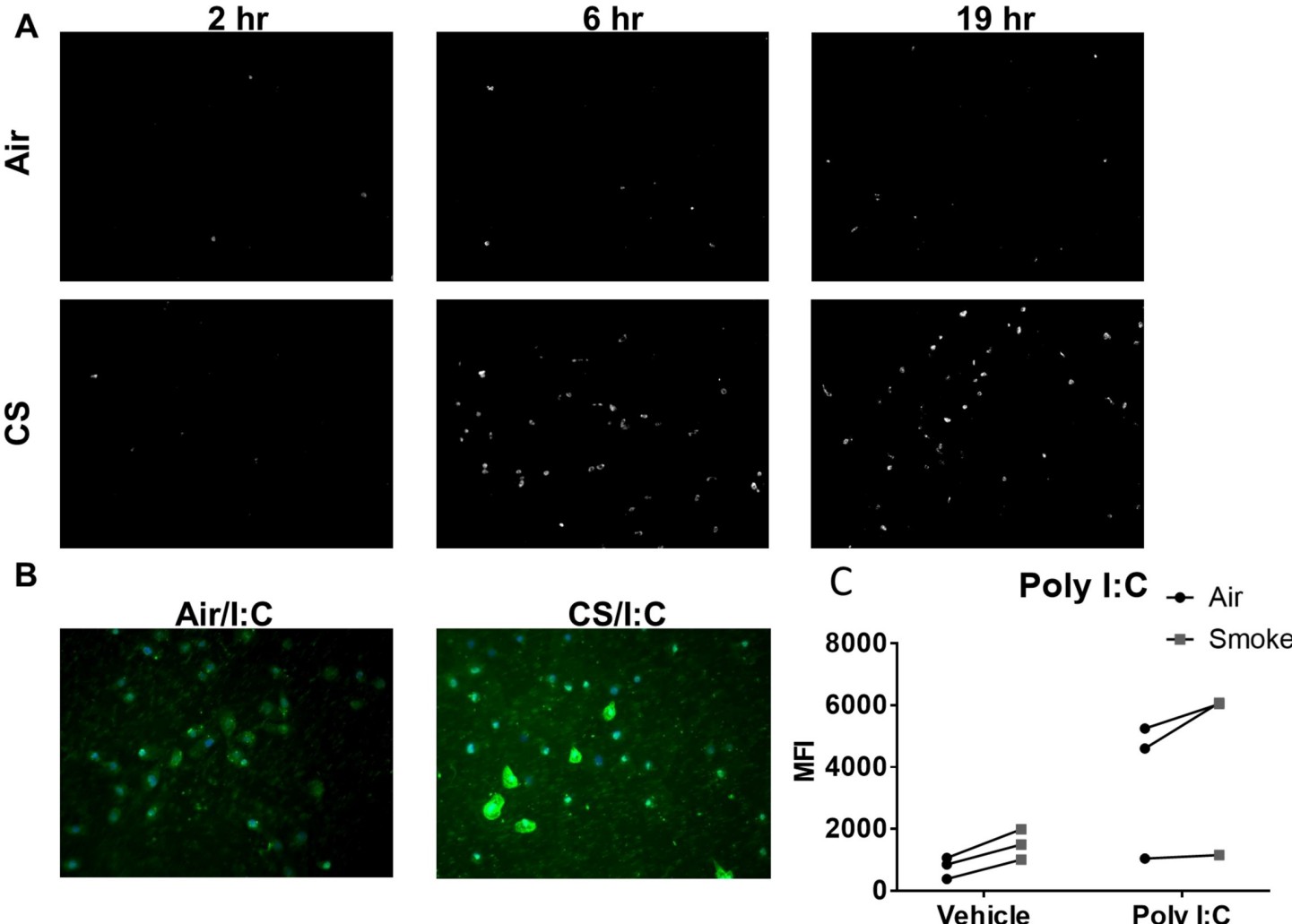

**Fig 2. Cigarette smoke increases uptake of poly I:C.** Primary human SAEC were exposed to either air or cigarette smoke for 60 minutes at the air-liquid interface. Following exposure cells were treated with 2.5μg/mL of fluorescent poly I:C for the indicated times. (A) Time course analysis of fluorescence at 2, 6, and 19 hours post poly I:C treatment. (B) Representative fluorescent images of poly I:C positive cells taken 6 hours post poly I:C treatment. (C) Analysis of cell fluorescence by flow cytometry 6 hours post treatment with poly I:C. The experiment was repeated using SAEC derived from 3 separate donors and each pair of points represents an SAEC strain from a different donor.

investigated the effect of cigarette smoke on macropinocytosis. Air or smoke-exposed SAEC were treated with FITC-Dextran, a common marker of macropinocytosis [24], for 6 hours. Similar to BSA, most air exposed cells took up FITC-dextran (Fig 5A). Cigarette smoke-exposed SAEC had decreased FITC-Dextran uptake (Fig 5A). We confirmed the effect of smoke-exposure on FITC-dextran uptake in SAEC using flow cytometry. We found that smoke-exposed SAEC had significantly reduced MFI compared to air exposed cells (Fig 5B). This was confirmed by decreased MFI in smoke-exposed cells treated with FITC-Dextran compared to air exposed cells treated with FITC-Dextran (Fig 5B). Taken together, our data shows that cigarette smoke selectively increases uptake of ligands by CavME while impairing CME and macropinocytosis.

One simple explanation for the changes in endocytosis would be if cigarette smoke changed the protein levels of caveolin or clathrin. Smoke or air exposed SAEC were harvested either

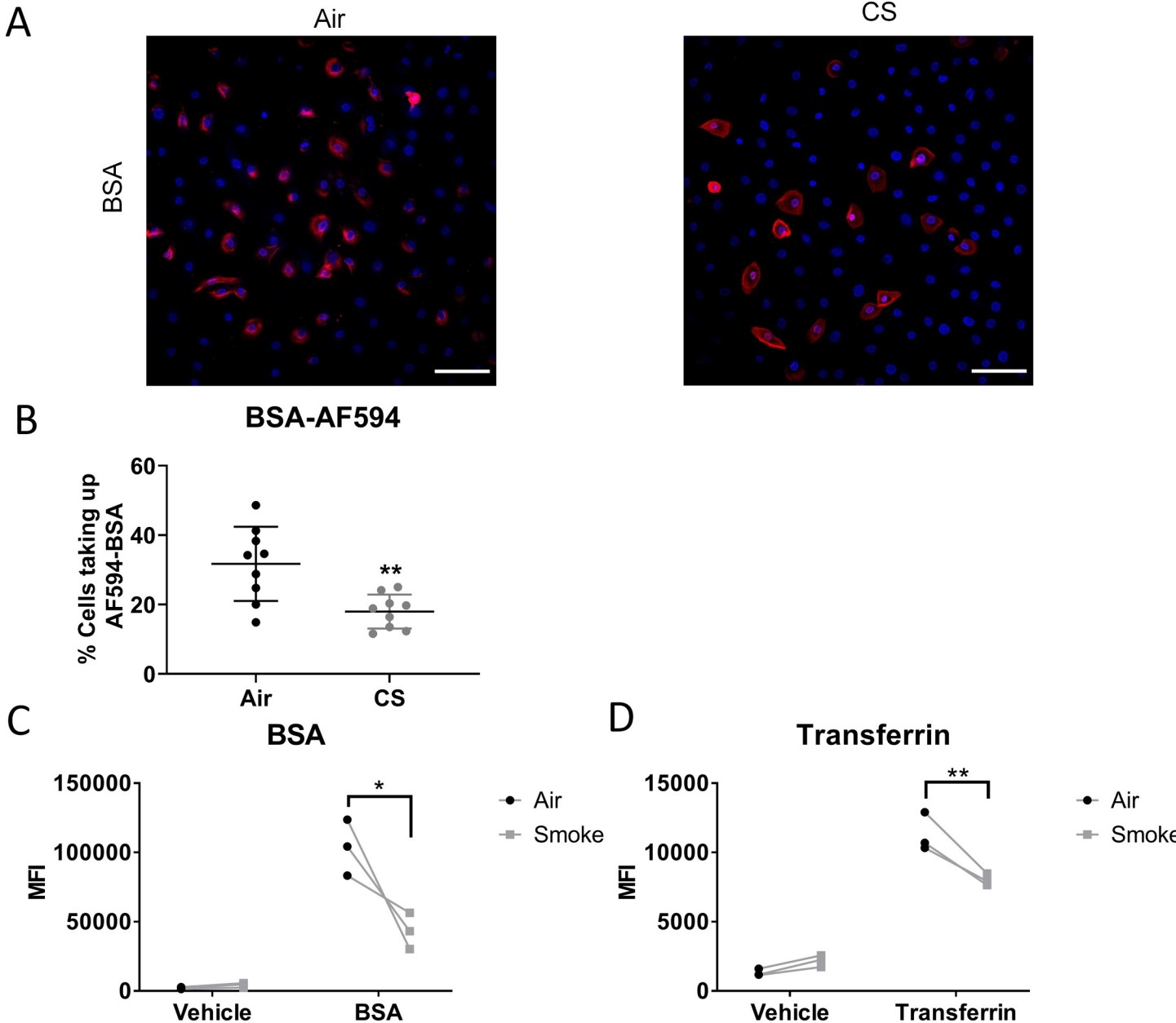

**Fig 3. Cigarette smoke decreases clathrin mediated endocytosis.** Primary human SAEC were exposed to either air or cigarette smoke for 60 minutes at the air-liquid interface. Following exposure cells were treated with AF594-labeled BSA (pictures in A, B) or AF488-labeled BSA (flow cytometry, C) (50μg/mL) or AF488-labeled transferrin (Tfn, 125μg/mL) as described in the materials and methods. (A) Photographs of air and smoke-exposed cells treated with AF594-BSA. Similar results were seen in 3 strains of SAEC tested. (B) AF594-BSA labeled cells were quantified as described in the Methods, using 3 high power fields per culture and three replicate cultures of one SAEC cell strain. Uptake of (C) AF488-BSA and (D) AF488-Tfn fluorescence was determined by flow cytometry. Each pair of points represents an SAEC strain from a different donor. *p<0.05 ***p<0.001 by two way ANOVA with Sidak correction.

immediately following or 6 hours post-smoke exposure and levels of clathrin and caveolin were detected by western blot. We found that protein levels of clathrin and caveolin were unchanged in whole cell lysate regardless of smoke exposure either immediately after or 6 hours after smoke exposure (Fig 6). This suggests that the effects of uptake in smoke-exposed SAEC is not due to changes in levels of clathrin or caveolin.

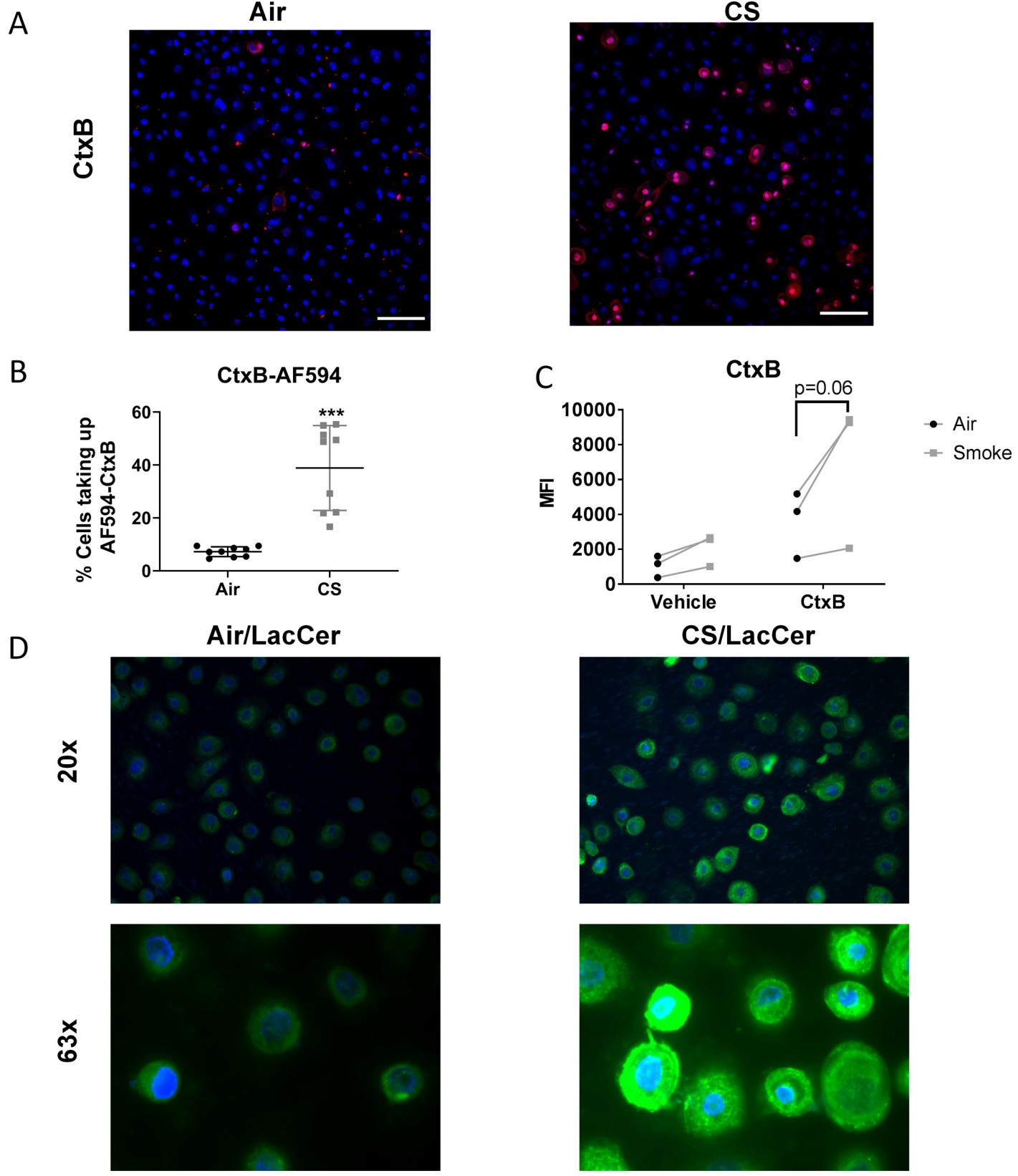

**Fig 4. Cigarette smoke increases caveolin mediated endocytosis.** Primary human SAEC were exposed to either air or cigarette smoke for 60 minutes at the air-liquid interface. (A) Following exposure cells were rested overnight and treated with AF594-labeled CtxB (10μg/mL) for 30 min and fixed for confocal microscopy. or BODIPY-LacCer (1μM) as described in the materials and methods. (B) Percentage of cells taking up AF594-CtxB was quantified as described in the Methods, using 3 high power fields per culture and three replicate cultures of one SAEC cell strain. (C) SAEC exposed to air or smoke were rested for 5 hr then treated with AF488-CtxB (10μg/ml) as described in materials and methods and were analyzed by flow cytometry. Each pair of points represents an SAEC strain from a different donor. ***p<0.001 by two way ANOVA with Sidak correction. (D) Photographs of air and smoke-exposed cells treated with BODIPY-LacCer.

Since many respiratory viruses, including IAV, utilize endocytosis for uptake into the host cell, we wanted to investigate the smoke-induced alterations of endocytosis could impact infection with a respiratory virus. To do this, air or smoke-exposed SAEC were rested for 5 hours and then infected with 0.01 MOI of the WSN strain of IAV. After 24 h, the total number of

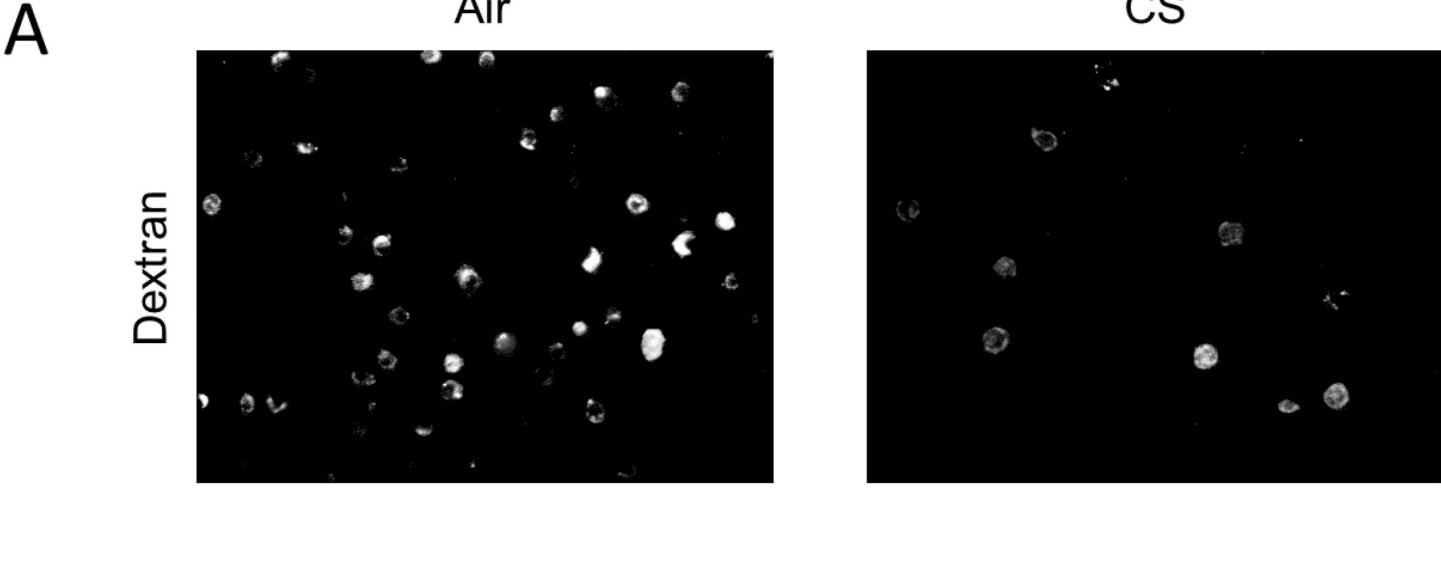

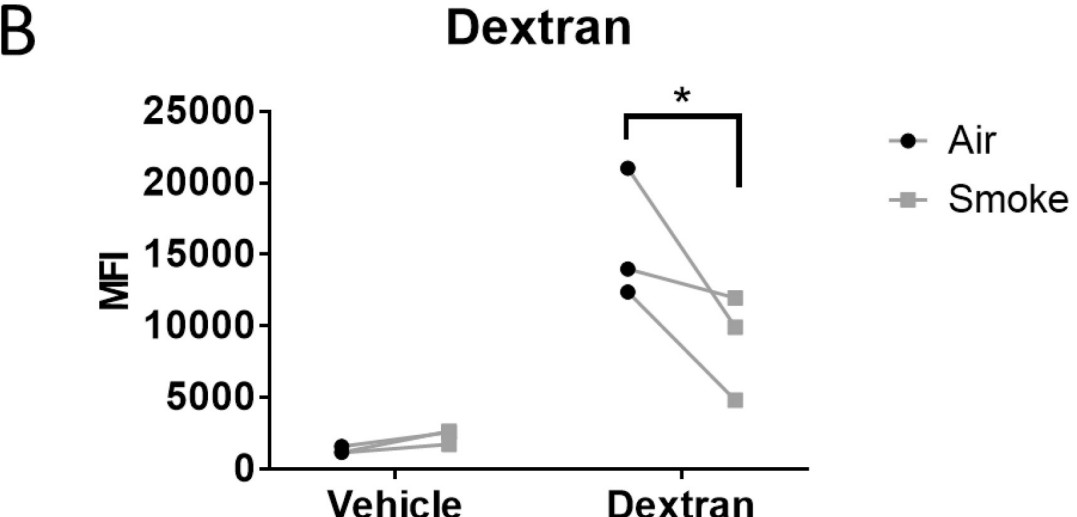

**Fig 5. Cigarette smoke decreases macropinoctosis.** Primary human SAEC were exposed to either air or cigarette smoke for 60 minutes at the air-liquid interface. Following exposure cells were treated with FITC-dextran (250 μg/mL) as described in the materials and methods. (A) Representative photographs from one cell strain. All three SAEC strains showed similar results. (B) Analysis of cell fluorescence by flow cytometry for FITC-dextran uptake. Each pair of points represents an SAEC strain from a different donor. *p<0.05 by two way ANOVA with Sidak correction.

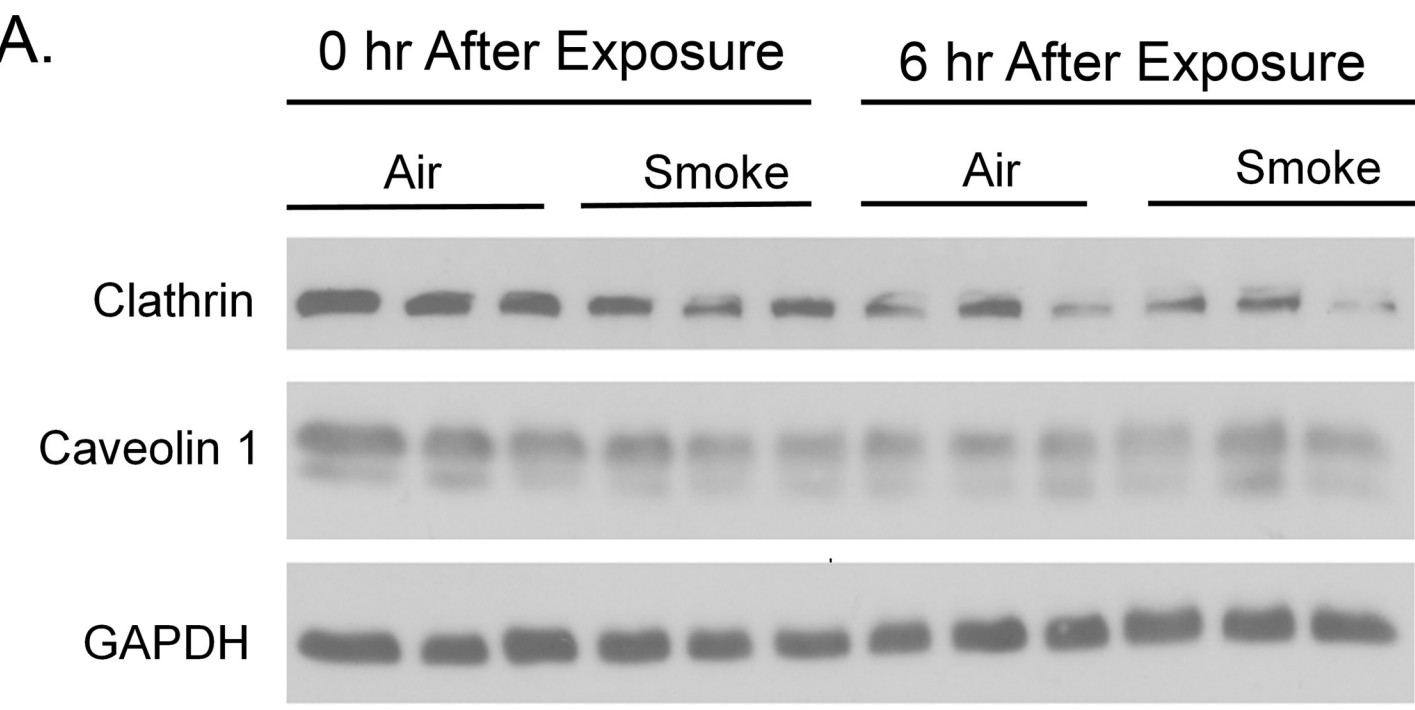

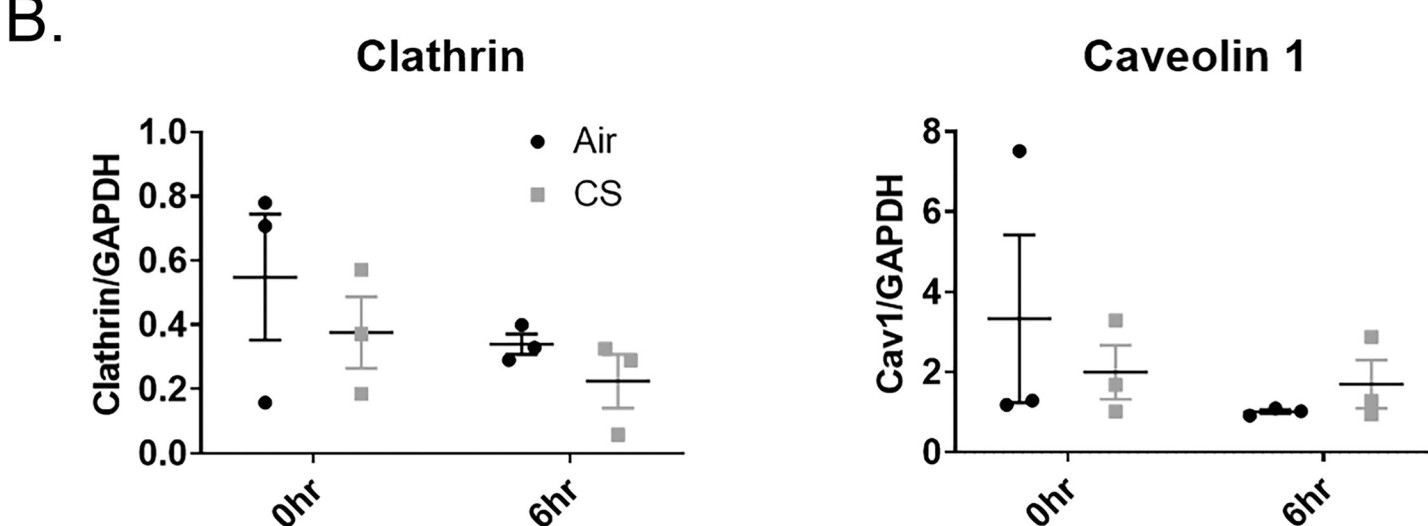

**Fig 6. Cigarette smoke does not alter protein levels of clathrin or caveolin.** (A) Primary human SAEC were exposed to either air or cigarette smoke for 60 minutes at the air-liquid interface. Cells were lysed immediately following smoke exposure or 6 hours post-smoke exposure and whole-cell levels of clathrin and caveolin were determined by Western Blot. (B) Caveolin and clathrin expression was quantified using densitometry normalized to GAPDH. No significant differences in expression were identified.

infected cells was determined by immunofluorescent staining for the NP protein. Similar to poly I:C and the caveolin pathway, smoke-exposed SAECs had an increased viral protein expression indicating increased infectivity and propagation (Fig 7A). IAV has been shown in many susceptible lines to be taken up by CME which is in contrast to our finding of increased

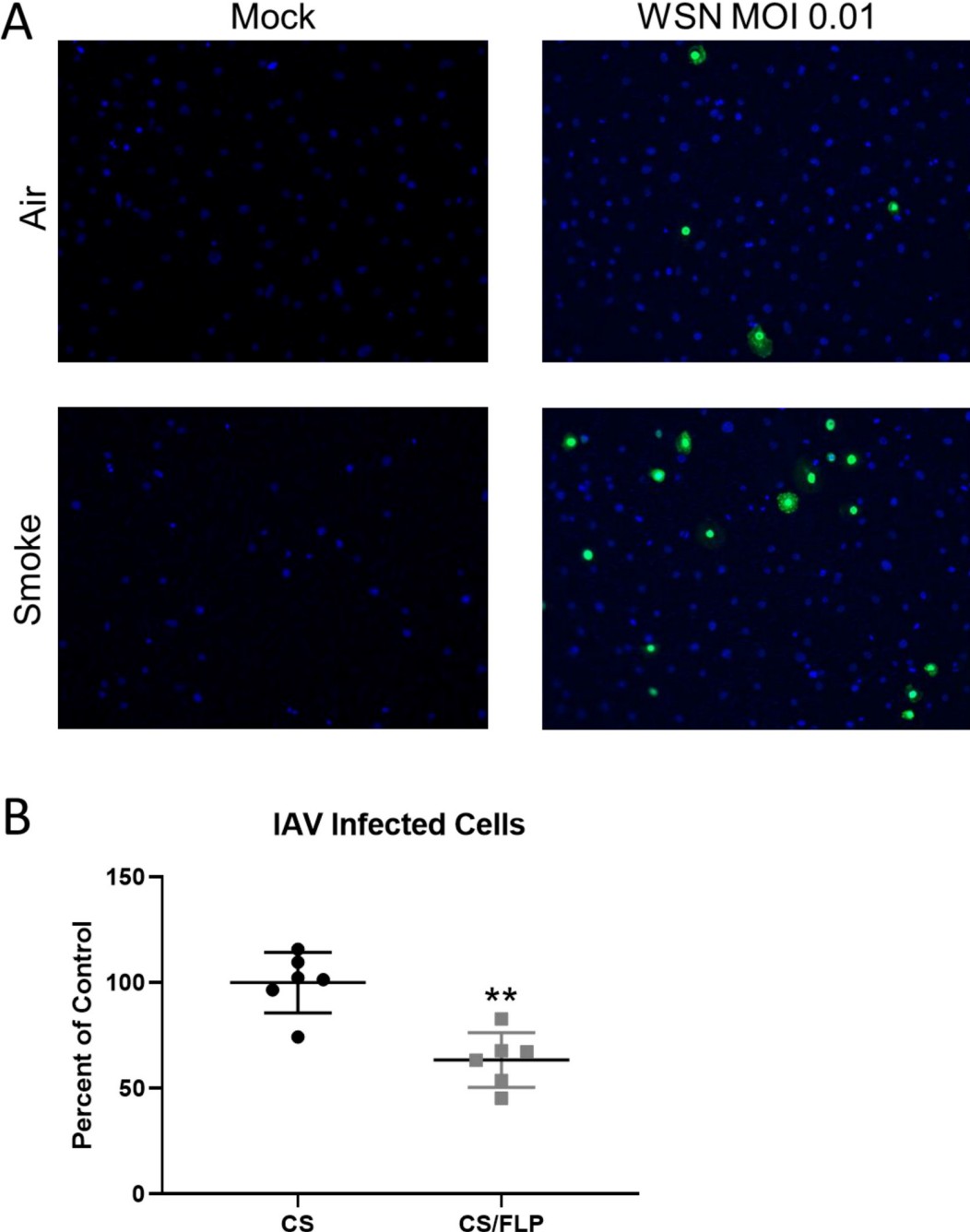

**Fig 7. Smoke increases infectivity of SAEC with IAV.** Primary human SAEC were exposed to either air or cigarette smoke for 60 minutes at the air-liquid interface. (A) Cells were then infected with WSN IAV at an MOI of 0.01 and fixed 24 hours post-infection and stained for IAV NP. (B) Smoke-exposed SAEC were treated with filipinIII or vehicle prior to infection with MOI 0.1 of IAV. 24 hours after infection, cells were fixed, stained, and imaged for IAV NP and the percent of NP positive cells was counted.

infectivity and decreased CME. We therefore hypothesized that smoke-exposed SAEC can utilize CavME to uptake IAV. To test this we infected smoke-exposed SAEC with a higher titer of IAV, to allow more detectable IAV infection, in the presence or absence of flipinIII, an

inhibitor of CavME. We found that inhibition of CavME in smoke-exposed SAEC resulted in reduced infectivity of SAEC (Fig 7B).

## Discussion

Cigarette smokers have increased incidence of viral infections in the lung [2, 3]. This is of particular concern to those with underlying lung diseases, such as COPD, as viral infections can trigger acute worsening of respiratory symptoms, resulting in disease progression and mortality [25–27]. While we and others have shown that antiviral signaling, including the production of key antiviral and inflammatory mediators, are impaired in smoke-exposed epithelial cells [4, 28, 29], the effect of cigarette smoke on epithelial cell physiology remains poorly understood.

Endocytosis is a critical cell function that allows cells to uptake nutrients and is critical to proper cell signaling [19]. Endocytosis also allows cells to sense the extracellular environment and respond to pathogens. Fittingly, many pathogens hijack endocytic pathways to gain entry into the host cell [30–32]. Thus, alteration in proper endocytic function can have detrimental consequences. Indeed, alterations of endocytosis are seen in many lung diseases, including cancer and pulmonary fibrosis [33–36]. We and others have shown that cigarette smoke impairs bacterial phagocytosis in macrophages [37–39], yet here, we are the first to show that cigarette smoke selected upregulates CavME while downregulating multiple other endocytosis pathways in primary human lung small airway epithelial cells.

Epithelial cells are the first line of defense against inhaled environmental insults and are also targeted by many respiratory viruses. Here, we show that cigarette smoke exposure causes increased uptake of the viral mimetic poly I:C (Fig 2). Despite this increased uptake, smoke-exposed epithelial cells have impaired production of inflammatory and antiviral mediators (Fig 1). While this seems contradictory, we have previously shown that TLR3 cleavage is disrupted in smoke-exposed cells corresponding with impaired antiviral signaling [4]. Thus, the impairment in TLR3 signaling can account for the decreased antiviral signaling despite increased TLR3 uptake.

Interestingly, it has been reported that poly I:C signaling is dependent on CME in monocyte-derived dendritic cells and in HeLa cells [40, 41]. Despite this, uptake of poly I:C in primary human lung cells has not been well studied and may utilize different or multiple pathways. The impact of cigarette smoke on endocytic pathways is not known though existing data suggests that cells can utilize different endocytic pathways depending on extracellular cues. For example, the epidermal growth factor receptor (EGFR), traditionally taken up by CME, can co-localize with caveolin in the presence of oxidative stress [42].

In order to identify the effect of cigarette smoke on different endocytic pathways, we followed the uptake of different fluorescent ligands. BSA has been shown to be taken up by CME in an immortalized lung cell line [20]. Similarly, BSA has been shown to be taken up through CME in human neutrophils, placenta and differentiated THP1 cells [43–45]. Tfn is another CME marker as uptake of the Tfn receptor requires CME [46]. We found that, unlike poly I:C, the uptake of both BSA and Tfn was decreased in smoke-exposed cells (Fig 3). This agrees with data showing treatment with hydrogen peroxide can reduce the uptake of CME markers [47, 48]. Similarly, Tfn receptor internalization is suppressed in the presence of oxidative stress [49]. Cigarette smoke is known to upregulate markers of oxidative stress and activate antioxidant signaling pathways, thus, reactive oxygen species production may be involved in the alteration of endocytosis [50].

CavME requires the formation of caveolae, and unlike CME, can carry larger cargo [51]. Here we show that uptake of two markers of CavME, CtxB and LacCer [16, 52], were increased

in smoke-exposed SAEC (Fig 4). CavME is dependent on membrane cholesterol to form lipid rafts as the base to form caveolae [53]. While the effect of cigarette smoke on membrane cholesterol levels is unknown, cigarette smoke is known to oxidize lipids in the membrane [54] which may affect membrane fluidity or caveolae formation.

Endocytosis can also occur in the absence of caveolin and clathrin. Indeed, many pathways have been identified that function independent of CME and CavME [55]. Macropinocytosis is one such pathway and allows for the uptake of soluble substrates from the extracellular space. FITC-Dextran is often used as a marker of macropinocytosis as it does not have a known membrane receptor [24]. We found that cigarette smoke-exposed SAEC had decreased uptake of FITC-Dextran (Fig 5). This agrees with another study showing that macrophage derived microvessicles, which were taken up by a mechanism consistent with fluid phase endocytosis, was decreased following treatment with smoke extract [56]. Taken together, our data shows that smoke-exposed cells preferentially increase uptake via CavME while decreasing uptake via CME and macropinocytosis. Our data is intriguing as it shows that cigarette smoke does not simply upregulate the uptake of all substrates, but selectively affects specific endocytosis pathways.

Since the effect of cigarette smoke was not the same on all endocytic pathways and seemed to favor CavME, we investigated the impact of cigarette smoke on proteins involved in endocytosis. Endocytosis can be regulated by altering the protein levels of adaptor molecules like caveolin and clathrin. Caveolin levels are decreased in many lung cancer cell lines which corresponds to altered adherence [57, 58] and increased caveolin levels corresponds to tumor drug resistance [59]. We measured the protein levels of clathrin and caveolin in air and smoke-exposed SAEC. Ultimately, we found that protein levels of clathrin and caveolin were unchanged in whole cell lysates (Fig 6).This suggests that the altered endocytosis in smoke exposed SAEC is not due to a change in the availability of endocytic adaptor proteins.

We lastly wanted to determine if the smoke induced changes in endocytosis were relevant in the context of a respiratory pathogen that requires endocytosis to enter the cell using IAV as a prototypic respiratory virus. We found that smoke exposure causes increased infectivity with IAV (Fig 7A). Reports have seen that IAV can enter cells in both clathrin and caveolin dependent manners [60–62]. We believe that despite decreases in CME, IAV can utilize other endocytic pathways, including CavME, to enter SAEC. To block uptake via CavME, we treated SAEC with filipinIII, an inhibitor of CavME. FilipinIII has been shown to cause loss of caveolae [63, 64] and reduce the uptake of CavME ligands such as CtxB [22, 65, 66]. We show that filipinIII treatment reduced the infectivity with IAV in our smoke exposed SAEC (Fig 7B). Thus, the upregulation of CavME in smoke-exposed SAEC could explain increases in IAV infectivity.

The increase in uptake of viral particles could prove consequential in the smoke-exposed lung. We have previously reported that cigarette smoke impairs TLR3 cleavage and thus suppressing type I and type III IFN production in SAEC [4]. The combination of increased viral uptake and impaired antiviral defenses can result in increased viral spread (Fig 7) and more severe infection. This may also be true for other pathogens that require endocytosis to infect the host cell. Cigarette smoke has also been shown to impair macrophage phagocytosis, a key step in clearing bacteria and apoptotic cells, thus further impairing the resolution of pathogen infection [38]. The net result of altered endocytosis becomes delayed pathogen clearance and more severe infection. Our work offers novel data that shows that cellular uptake pathways are dysregulated in epithelial cells after cigarette smoke exposure which impacts specific ligands in smoke-exposed cells.

## Supporting information

**S1 Raw images.**
(PDF)

## Author Contributions

**Conceptualization:** Parker F. Duffney, Thomas H. Thatcher, Richard P. Phipps, Patricia J. Sime.

**Data curation:** Thomas H. Thatcher.

**Formal analysis:** Thomas H. Thatcher, Patricia J. Sime.

**Funding acquisition:** Richard P. Phipps, Patricia J. Sime.

**Investigation:** Parker F. Duffney, A. Karim Embong, Connor C. McGuire.

**Methodology:** Parker F. Duffney, Patricia J. Sime.

**Project administration:** Patricia J. Sime.

**Supervision:** Thomas H. Thatcher, Patricia J. Sime.

**Writing – original draft:** Parker F. Duffney.

**Writing – review & editing:** Parker F. Duffney, Thomas H. Thatcher, Richard P. Phipps, Patricia J. Sime.

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
