## [Decision Letter · Decision Letter 0]

5 Dec 2019

PONE-D-19-26455

Cigarette Smoke Increases Susceptibility to Infection in Lung Epithelial Cells by Upregulating Caveolin-Dependent Endocytosis

PLOS ONE

Dear Prof. Sime,

Dear Patricia,

Thank you for submitting your manuscript to PLOS ONE. After careful consideration, we feel that it has merit but does not fully meet PLOS ONE’s publication criteria as it currently stands. Therefore, we invite you to submit a revised version of the manuscript that addresses the points raised during the review process.

We would appreciate receiving your revised manuscript by Jan 19 2020 11:59PM. To enhance the reproducibility of your results, we recommend that if applicable you deposit your laboratory protocols in protocols.io, where a protocol can be assigned its own identifier (DOI) such that it can be cited independently in the future. For instructions see: http://journals.plos.org/plosone/s/submission-guidelines#loc-laboratory-protocols

We look forward to receiving your revised manuscript.

Kind regards,

Ali Önder Yildirim, PhD

Academic Editor

PLOS ONE

Journal Requirements:

3. Please note that all PLOS journals ask authors to adhere to our policies for sharing of data and materials: https://journals.plos.org/plosone/s/data-availability. According to PLOS ONE’s Data Availability policy, we require that the minimal dataset underlying results reported in the submission must be made immediately and freely available at the time of publication. As such, please remove any instances of 'unpublished data' or 'data not shown' in your manuscript and replace these with either the relevant data (in the form of additional figures, tables or descriptive text, as appropriate), a citation to where the data can be found, or remove altogether any statements supported by data not presented in the manuscript.

4. In your Methods section, please give the sources of the ISRE reporter cell line used in your study. Please also provide details for the ELISA reagents used to detect the cytokine panel in your study.

Reviewers' comments:

Reviewer's Responses to Questions

**Comments to the Author**

1. Is the manuscript technically sound, and do the data support the conclusions?

Reviewer #1: Partly

2. Has the statistical analysis been performed appropriately and rigorously? 

Reviewer #1: Yes

3. Have the authors made all data underlying the findings in their manuscript fully available?

Reviewer #1: Yes

4. Is the manuscript presented in an intelligible fashion and written in standard English?

Reviewer #1: Yes

5. Review Comments to the Author

Reviewer #1: Parker F Duffney and colleagues present a manuscript that investigates the mechanism by which lung epithelial cells exposed to CS get more susceptible to virus infections. They identified the caveolin-dependent endocytosis as the main path for viruses to enter epithelial cells of smokers. Some experiments need to be included to meet the perfect format and requirements for a fully acceptable work.

6. PLOS authors have the option to publish the peer review history of their article (what does this mean?). If published, this will include your full peer review and any attached files.

Reviewer #1: Yes: Dr. Aicha Jeridi

---

## [Author Response · Author response to Decision Letter 0]

21 Feb 2020

Response to reviewer comments

Figure 3: 

A. Include bright field to have a better clue where the labeled-BSA (BSA-AF594) is located in the cell after uptake. 

Authors do not mention how long cells were incubated with the dye and when the readout was (in the legend they refer to the methods part and in the methods part they say “cells were then treated ….for the indicated times”. Important to know whether authors, similar to the already published data on uptake of FITC-labeled BSA look at the cells after 60min or whether they wait longer. Fluorescent signal in reference nr. 20 shows a punctated localization which differs from the presented pictures here in Figure 3 (due to timing?)

We appreciate the reviewer comment. Unfortunately, we did not take brightfield images for of the cells in figure 3A. Despite this, we think the image sufficiently shows labeling consistent with both surface and intracellular staining. We updated the methods (lines 146-149) to clarify that for the confocal images, cells were loaded with ligand for 30 min at 4°C and then incubated at 37°C for 30 min before fixing and visualizing. While other studies have seen distinct puncta staining at 60 minutes, this shortened incubation time could explain why we see both surface and internal labeling. 

B. Specify the y-axis more – (%) BSA-uptake 

We have changed the axis to specify the uptake in Figure 3B is BSA uptake (New graph for figure 3B). 

Figure 4:

A. CtxB is also here a labeled one, please keep the same name in case of similar quantifications (CtxB-AF594 like in 4B), same for Figure C

We thank the reviewer for the comment and have adjusted the labeling for consistency throughout the figure (Line 237-242). The AF594-CtxB was used for the confocal imaging and subsequent image quantification. AF488-CtxB was used for flow cytometry labeling. We adjusted the labeling of Figures 4B and 4C to clarify this. 

D: the 63 magnification of the CS/LacCer picture shows a stronger 

 background signal than the one of the control. Users should take a 

 magnicication of a cell that is rather far away from the other positive cells 

 to avoid misunderstanding of exposure differences. 

Both images were acquired at the same exposure settings, and we adjusted the contrast to see clearly the label uptake in untreated cells. Because the smoke-exposed cells took up so much more label, this resulted in overexposed images of smoke-exposed cells. Unfortunately, these were semi-permanent mounts only and are no longer available to re-image. We have adjusted the contract on the images presented (using the same settings for both) to reduce the background, this unfortunately reduces the ability to see the label uptake in the control cells but we think the images will still be suitable for publication. 

Figure 5: 

Line 281/282: cellular fluorescence was difficult to quantify using automated image analysis… (First: there is no need to rely on only automated image analyzer, authors could optionally quantify single images, second: no need to mention this in the text since the picture is convincing). 

We appreciate the feedback from the reviewer. We removed the sentence mentioning the difficulties of image analysis with dextran and highlight that similar suppression of dextran uptake was seen when quantitating smoke exposed cells by flow cytometry (Line 525-524). 

Figure 6:

Authors should try to show the blots such that the bands are under each other. Labeling of the samples is not straight forward (immediate?), it would be rather better if authors used “short exposure” and “6h exposure”. In order to prove that there are no changes in clathrin and caveolin upon CS exposure, authors should add the quantification of band intensities. In addition, the second strain shows a slight reduction in both samples, the one immediately and the one 6h post CS exposure, how do authors deal with this. 

We have gone back and adjusted the western blot figure to line up the bands and make the labeling more clear for the reader. We have also performed densitometry on all the blots from all the strains and found no consistent or significant difference in the expression of clathrin or caveolin in our three strains as a whole. This data is included in new figure 6. While one strain did appear to show significantly decreased protein levels of caveolin (but not clathrin) in the smoke exposed cells, this would not explain why caveolin dependent uptake is enhanced in smoke exposed cells and this finding is not consistent in the other strains tested. Thus we feel that overall, the data does not support the idea that endocytosis changes due to smoke exposure are related to changes in the level of endocytosis proteins (either clathrin or caveolin).

Figure 7: very nice experiment, that shows the specificity of the virus uptake. It is necessary to exclude any effects of filipin III on the previously described pathways. Meaning, authors should include the experiments of labeled BSA and labeled CtxB with fillipin III and show that this does not affect the uptake of these compounds to finally confirm the uptake of the WSN virus exclusively by caveolin-dependent endocytosis. 

We agree with the reviewer that showing that showing the effect of filipin III on the uptake of BSA and CtxB would help to strengthen the findings of the figure. However, due to a recent relocation of the lab, the smoke exposure system is temporarily unavailable to perform additional experiments. However, there has been much research on the uptake of BSA and CtxB and it has been shown in numerous studies that BSA uptake utilizes a clathrin dependent mechanism (1-5). The inhibitor, filipin III, has been shown to disrupt caveole formation (6, 7) and CtxB is a common marker of lipid rafts (8-10) and has been shown to be inhibited by filipin III (11, 12). We have added additional information on this topic at lines 361 and 402-404. Despite the inability to perform additional experiments we believe this paper still holds value to the scientific community in its current form.

Notes:

Line 279: a common marker OF macropinocytosis…

This has been fixed.

---

## [Decision Letter · Decision Letter 1]

8 Apr 2020

Cigarette Smoke Increases Susceptibility to Infection in Lung Epithelial Cells by Upregulating Caveolin-Dependent Endocytosis

PONE-D-19-26455R1

Dear Dr. Sime,

We are pleased to inform you that your manuscript has been judged scientifically suitable for publication and will be formally accepted for publication once it complies with all outstanding technical requirements.

With kind regards,

Y. Peter Di, Ph.D.

Academic Editor

PLOS ONE

Additional Editor Comments (optional):

Reviewers' comments:

Reviewer's Responses to Questions

**Comments to the Author**

1. If the authors have adequately addressed your comments raised in a previous round of review and you feel that this manuscript is now acceptable for publication, you may indicate that here to bypass the “Comments to the Author” section, enter your conflict of interest statement in the “Confidential to Editor” section, and submit your "Accept" recommendation.

Reviewer #1: All comments have been addressed

2. Is the manuscript technically sound, and do the data support the conclusions?

Reviewer #1: Yes

3. Has the statistical analysis been performed appropriately and rigorously? 

Reviewer #1: Yes

4. Have the authors made all data underlying the findings in their manuscript fully available?

Reviewer #1: Yes

5. Is the manuscript presented in an intelligible fashion and written in standard English?

Reviewer #1: Yes

6. Review Comments to the Author

Reviewer #1: All the questions that I mentioned in my first review were answered and addressed in a proper way. Authors put a lot of effort in adding new data to round up the story. The publication is in my point of view, after the modifications done by authors, ready to be published. Many thanks

7. PLOS authors have the option to publish the peer review history of their article (what does this mean?). If published, this will include your full peer review and any attached files.

Reviewer #1: No

---

## [Editor Report · Acceptance letter]

4 May 2020

PONE-D-19-26455R1 

Cigarette Smoke Increases Susceptibility to Infection in Lung Epithelial Cells by Upregulating Caveolin-Dependent Endocytosis 

Dear Dr. Sime:

I am pleased to inform you that your manuscript has been deemed suitable for publication in PLOS ONE. Congratulations! Your manuscript is now with our production department. 

With kind regards,

on behalf of

Dr. Y. Peter Di 

Academic Editor

PLOS ONE